# Stacked Boosters Network Architecture for Short-Term Load Forecasting in Buildings

**Tuukka Salmi \*, Jussi Kiljander** 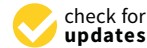 **and Daniel Pakkala**

VTT Technical Research Centre of Finland, FI-02044 Espoo, Finland; jussi.kiljander@vtt.fi (J.K.);
daniel.pakkala@vtt.fi (D.P.)
\* Correspondence: tuukka.salmi@vtt.fi

**Abstract:** This paper presents a novel deep learning architecture for short-term load forecasting of building energy loads. The architecture is based on a simple base learner and multiple boosting systems that are modelled as a single deep neural network. The architecture transforms the original multivariate time series into multiple cascading univariate time series. Together with sparse interactions, parameter sharing and equivariant representations, this approach makes it possible to combat against overfitting while still achieving good presentation power with a deep network architecture. The architecture is evaluated in several short-term load forecasting tasks with energy data from an office building in Finland. The proposed architecture outperforms state-of-the-art load forecasting model in all the tasks.

**Keywords:** deep neural networks; short-term load forecasting

## 1. Introduction

Due to increasing utilization of variable renewable energy (VAE), demand flexibility is becoming crucial part of the stabilization of smart grids. Buildings constitute 32% of global final energy consumption [1] and possess natural thermal storage capacity making them important resource for demand flexibility. Accurate short-term load forecasting is crucial part of demand-side flexibility management.

Short-term energy load forecasting has been studied extensively, but there are some limitations in the existing state-of-the-art solutions. According to a recent study by Amasyali et al. [2], Artificial Neural Networks (ANNs) are the most commonly used machine learning model in short-term forecasting of building energy loads. The challenge with artificial neural networks is that they require a lot of data to prevent overfitting. This is usually tackled by reducing the number of layers, which in turn reduces the presentation power of the model. For instance, a very typical neural network architecture in building load forecasting is a single hidden layer multilayer perceptron (MLP) model [3–6]. Besides neural network methods, support vector regression (SVR) is a common machine learning model for building load forecasting [7–9]. Although SVR is not a neural network model, it can be though as a single hidden layer ANN from the point view of the model capacity.

In addition to the shallow MLP and SVR models, deeper model architectures for building load forecasting have been proposed. Marino et al. [10] compare standard long short-term memory (LSTM) and LSTM sequence-to-sequence architectures (also known as encode-decoder) for building energy load forecasting. LSTM based architecture is also studied by Kong et al. in two papers [11,12]. Although the LSTM architectures studied in all three papers consist of only two stacked LSTM layers, the recurrent neural networks (RNN) such as LSTM are deep models due to the recursive call made at every time step. Amarasinghe et al. [13] present study on a convolutional neural networks (CNN) model for building load forecasting. The model consist of three CNN layers and two fully connected

layers. Yan et al. [14] propose a CNN-LSTM network consisting of two CNN layers and a single LSTM layer. The model is evaluated in five different building data sets. Mocanu et al. [15] study the performance of conditional restricted Boltzmann machine (CRBM) and factored conditional restricted Boltzmann machine (FCRBM) neural network architectures in short-term load forecasting.

A very deep ANN architecture for energy load forecasting is presented by Chen et al. [16]. The authors adopt residual connections [17], a successful approach for building deep CNNs for image processing, and form a 60 layer deep network for energy load forecasting. The proposed ResNet borrows some ideas from RNN type network since it contains connections from previous hour forecast to next hour forecasts, but the connection is not an actual feedback since there are no parameter sharing among different hour forecasters. The proposed ResNet combined with an ensemble approach achieves state of the art results in three public energy forecasting benchmarks. Although none of these data sets focuses on individual building loads, the proposed work is general energy load forecasting model and thus good state-of-the-art benchmark also for building level forecasting tasks.

The aforementioned deep learning models have been evaluated using data sets with large amount of relatively static training data, which allows the models to avoid overfitting and outperform shallower models. However, in many situations (e.g., with new buildings or locations with extreme weather conditions) it is useful to be able to avoid overfitting also in situations with limited amount of training data. Moreover, a big limitation of the machine learning methods in general is that they assume that the data distribution does not change. This assumption can cause problems when machine learning methods such as neural networks or SVR are used for modelling real-life systems that change over time. In the context of buildings these changes in the data distribution can be caused, for example, by changes in the people (e.g., habits of people change over time or new people move in) or in the building infrastructure (e.g., new appliances, or configuration of existing systems such as HVAC change). A good example on how the data distributions can change dramatically is demonstrated by the Covid-19 pandemic, which has reduced usage rate of office building dramatically and therefore affected to energy consumption of these buildings.

We propose a novel hierarchical neural network architecture for short-term load forecasting that has been designed to address the abovementioned limitations of state-of-the-art load forecasting models. The architecture, called Stacked Booster Network (SBN), tries to achieve the good properties of deep models while still avoiding overfitting in small sample size situations. The core idea is to reduce the model parameter space with following principles: (1) sparse interactions, (2) parameter sharing, and (3) equivariant representations. Another key idea of the architecture is a novel boosting technique, which makes it possible to transform the original multivariate time series problem into univariate one. This further reduces the model parameters while keeping the network capacity high enough. Additionally, the proposed boosting technique enables the model to correct systematic mistakes by utilizing residual information on historical forecasts. With these ideas we can build a deep learning framework for short-term load forecasting with following properties:

- Minimal number of parameters leading to robust training even with small amount of training data
- Sufficiently large number of layers leading to enough presentation power for modelling real phenomenon of the modelled load
- Boosting technique that allows the model to adapt to changing data distributions, which are typical in real-life data

The paper is organized as follows. In Section 2, a general network architecture of the SBN is presented. Section 3 introduces the case studies and presents the evaluation where the SBN is compared to the state-of-the-art ResNet model [16]. Section 4 concludes the presentation and presents directions for future work.

## 2. SBN Architecture

The SBN is a general neural network architecture style, which can be instantiated in different ways. This section presents the general SBN architecture style and its main ideas and design principles. A concrete instance of the SBN for a particular case study with associated implementation details is presented in Section 3.3.

General model architecture is described in Figure 1. The architecture is composed of forecasting submodels at four levels:

- *Instant forecaster*
- *Hourly boosting forecaster*
- *Daily boosting forecaster*
- *Weekly boosting forecaster*

The architecture and configuration of the Instant forecaster depends on the energy system to be modelled. Typically, it takes as input a multivariate time series consisting of some combination of weather, temporal and energy features. A key idea in the SBN architecture, is that the *Instant forecaster* is the only part of the model that needs to manipulate the multivariate timeseries. The boosting forecasters are more general and designed to be stacked over previous forecaster in a way that the next forecaster boosts the previous. To this end, each booster gets a timeseries of past residuals as input and forecast the error made by the previous layer of models.

The architecture of the whole boosting system is shown in Figure 1 while the architecture of an individual booster is shown Figure 2. Architecture of the *Instant forecaster* is shown in Figure 3.

The next three subsections cover each of the forecasters with more details with their limitations. Then, we present a more detailed demonstration on how the SBN works with illustrative figures. Final subsection covers some general implementation notes of the SBN.

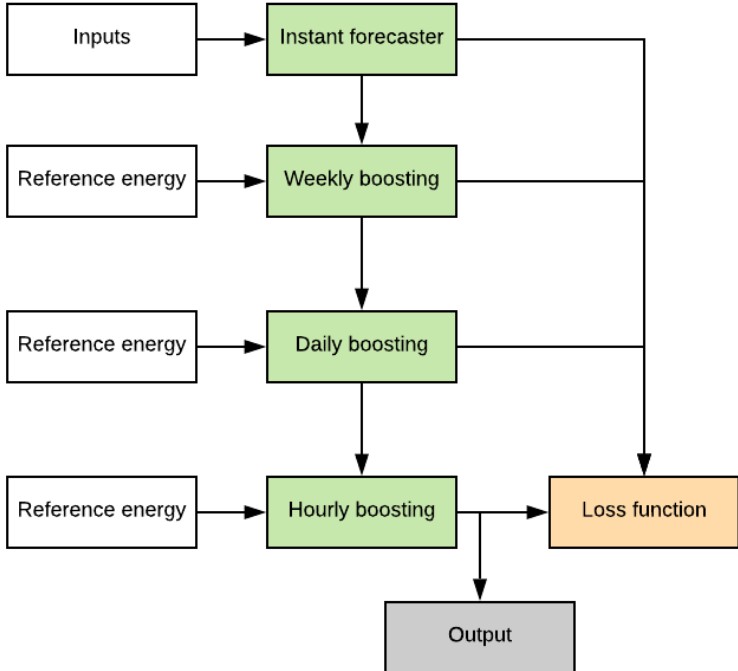

**Figure 1.** SBN architecture.

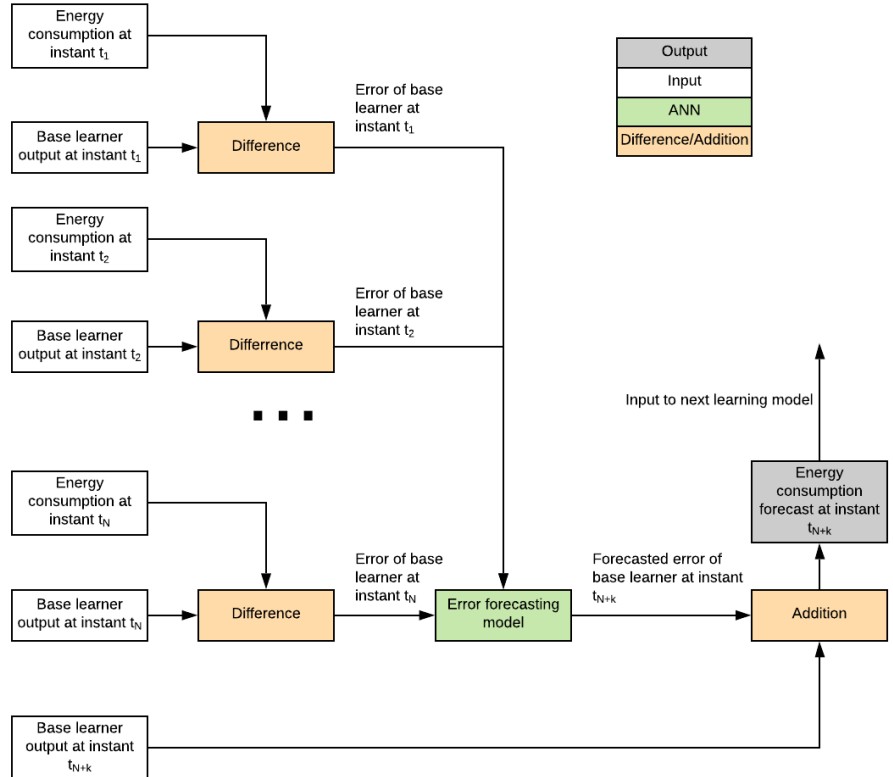

**Figure 2.** Architecture of a single boosting forecasters. Each boosting forecaster has the same general architecture. Base learner in the figure is previous booster or the instant forecaster in case of the first booster. In the figure, value $N$ is the amount of previous errors used in forecasting future error and value $k$ is the amount of time steps to be forecasted.

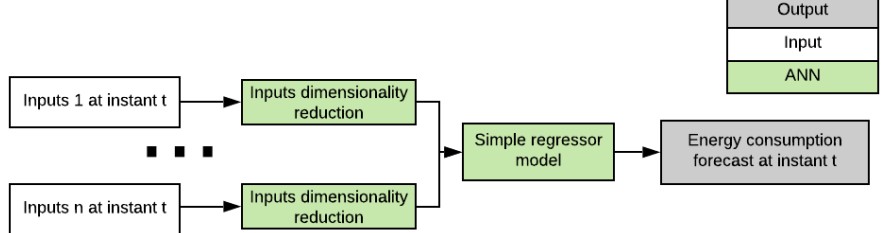

**Figure 3.** Architecture of the *Instant forecaster*.

### 2.1. Instant Forecaster

The purpose of the *Instant forecaster* is to forecast energy consumption based on independent variables that are assumed to be fully observable without any information on past energy consumptions. In our case study, described with details in Section 3, the only independent time series are temperature, hour of the day, and day of the week. Here also temperature can be considered fully observable by replacing future temperature values by their forecasts. These independent variables are first feed with dimensionality reduction submodels to reduce time dimension of these univariate time series to one. Considering our case study, hour of the day and day of the week are deterministic time series and do not therefore require dimensionality reduction submodel.

It should be noted that we make a reasonable assumption that different temperature profiles behave exactly same way at every weekday and every hour. Therefore, with this assumption reducing temperature sequence dimension to one before merging to other inputs does not lose any usable information. If some other information sources would be used, it is crucial to analyse whether the approach loses information and whether extra complexity needs to be addressed in the submodel.

### 2.2. Weekly Boosting Forecaster

Energy time series have typically periodical loads that occur same time on specific days of the week. If the loads would be always at the same level at the same time, the *Instant forecaster* would estimate them perfectly. However, it is quite typical that the timing and volume of these loads varies over time. This is the phenomena that the *Weekly boosting forecaster* should compensate. Consider, for example, a load occurring at some specific weekday rapidly increases by fixed amount. Then the *Weekly boosting forecaster* gets time series having this constant value as an input, and forecasts this constant value for the future errors. Therefore, in this case, the *Weekly boosting forecaster* is capable for estimating this dynamic change perfectly after short reaction time.

### 2.3. Daily Boosting Forecaster

The *Daily boosting forecaster* works exactly same way as the *Weekly boosting forecaster* but it focuses on changes on loads that occurs daily basis rather than weekly. Consider, for example that daily occurring load rapidly increases by fixed amount. The *Weekly boosting forecaster* would need weeks of data to be able to estimate and correct it. Therefore, it remains uncorrected at the beginning. The role of the *Daily boosting forecaster* is address this delay.

### 2.4. Hourly Boosting Forecaster

Some energy time series have periodic fluctuations in the energy consumption that cannot be explained purely by the input data. This can be explained by phenomena where some devices may be turned on and off periodically and this periodicity can be changing by some unknown conditions. Therefore, this phenomena can be seen as error signal of the *Instant forecaster*. Moreover, if the periodicity is not divisible by 24 h, the *Weekly boosting forecaster* and the *Daily boosting forecaster* cannot compensate it. This is the phenomena that the *Hourly boosting forecaster* should compensate. Another more minor phenomena that hourly boosting forecaster should compensate is a situation where a constantly occurring load changes somehow. In this case weekly and daily boosting forecasters react with days delays but the *Hourly boosting forecaster* can react and estimate the errors more rapidly.

### 2.5. Demonstration of Operation

This section demonstrates the SBN network operation with figures. To simplify demonstration, the *Hourly boosting forecaster* is omitted. In this demonstration, the *Weekly Booster forecaster* uses three weeks of data and the *Daily booster forecaster* uses seven days of data.

In this setup, we consider forecasting some fixed hour on Day 1 on Week 4. First, we run the *Instant forecaster* for all the historical values (Figure 4). Then, we calculate the errors for these historical values using the real historical knowledge of energy consumption (Figure 5). Next step is to use the *Weekly boosting forecaster* to forecast error of Week 3 for each day by using errors of *Instant forecaster* on Weeks 1 and 2 (Figure 6). We correct errors of the *Instant forecaster* for Week 3 by estimated errors of the *Weekly booster forecasters* for Week 3. This way we get seven weekly boosted errors for Week 3. By using these values we forecast the error of our final forecasted time on Day 1 on Week 4.

Now, we are ready for forecasting energy consumption for our target time on Day 1 on Week 4. This step is demonstrated in Figure 7. Here we first run the *Instant forecaster* on the target day. Then we subtract the estimated weekly boosted error from this instant forecast. Finally, we subtract daily boosted forecast error from the weekly boosted forecast to get the final forecast.

From this illustration, one should note that the *Weekly boosting forecaster* uses two data points for forecasting the third and the *Daily boosting forecaster* uses seven data points for forecasting the eight. In more general, the last boosting forecaster uses the whole input length for making forecast while the previous boosting forecasters must use one data point less.

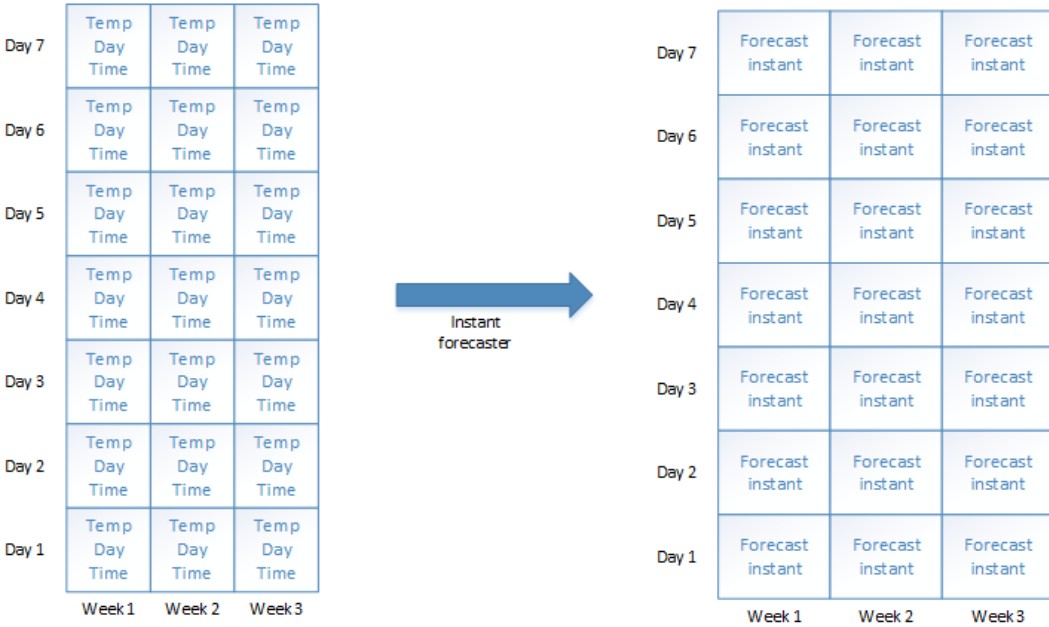

**Figure 4.** Step 1: Run historical values for given window through the *Instant forecaster*.

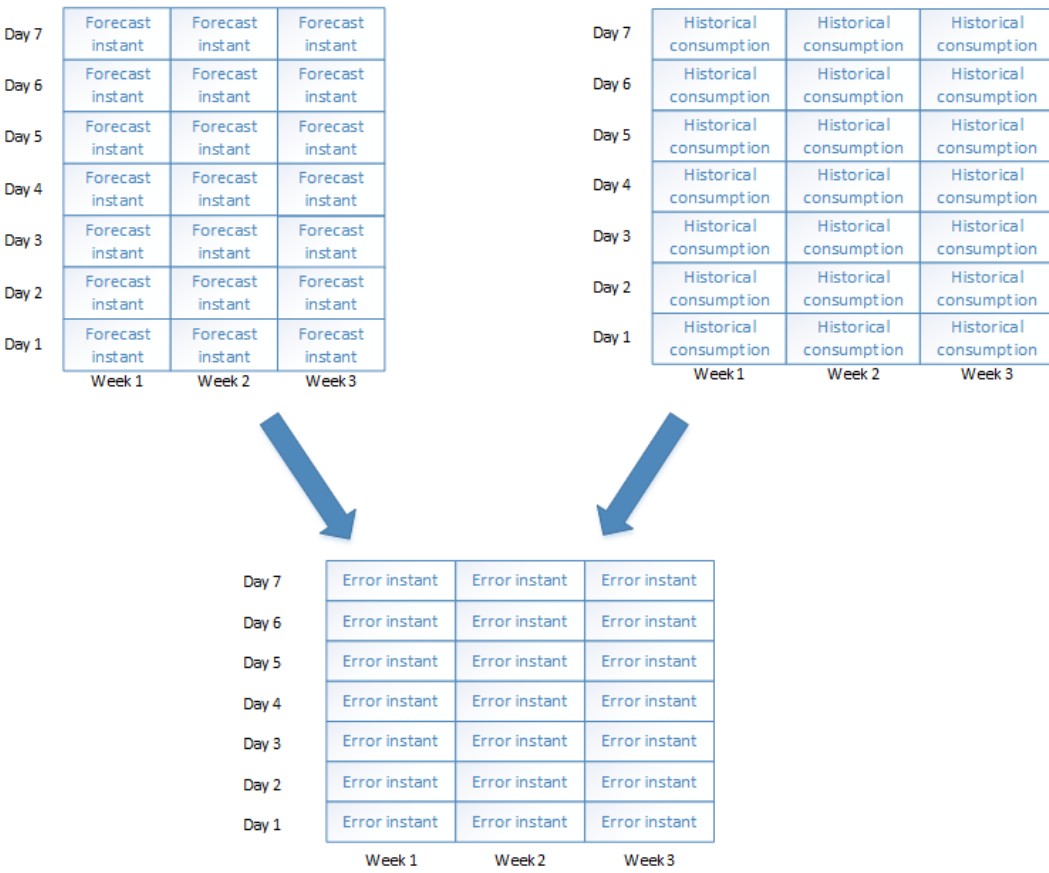

**Figure 5.** Step 2: Calculate errors of the *Instant forecasters* for historical data.

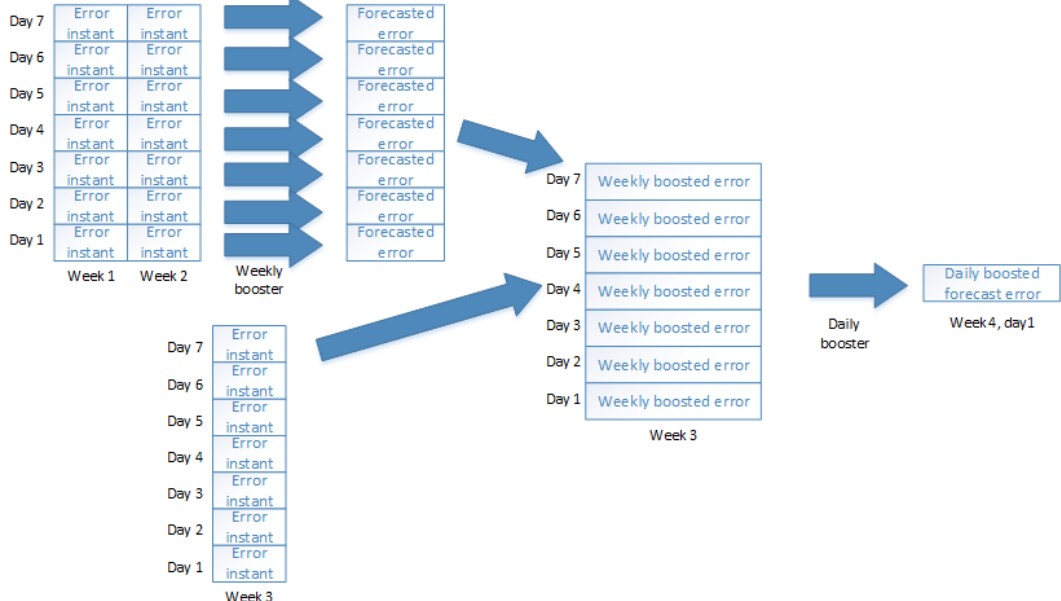

**Figure 6.** Step 3: Run historical values through the *Weekly boosting forecaster* and the *Daily boosting forecaster*.

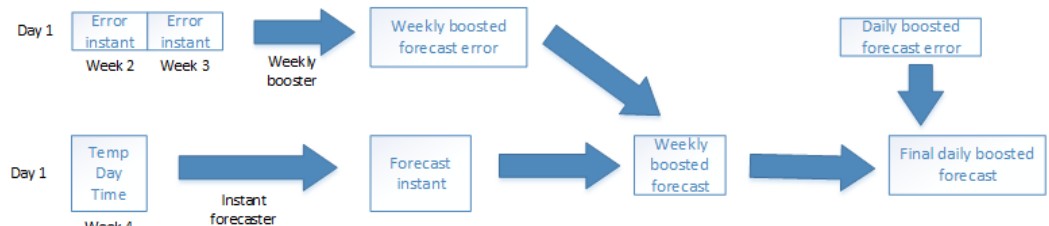

**Figure 7.** Make the final forecast using the *Instant forecaster* and compensate the forecast with the *Weekly boosting forecaster* and the *Daily boosting forecaster*.

*2.6. Implementation Notes*

It should be noted that stacking of the *Hourly boosting forecaster*, the *Daily boosting forecaster* and the *Weekly boosting forecaster* can be done in any order. Moreover, some of the forecasters may not be needed at all. In fact, higher time resolution boosting forecaster can correct exactly the same issues than the lower time resolution forecaster when the time window it gets as input is big enough. However, due to fact that some energy load occurs typically in daily and weekly basis, these energy loads are more easily estimated and compensated by the *Daily boosting forecaster* and the *Weekly boosting forecaster*. We have not experimented different ordering of the boosters and our experiments presented in Section 3 uses only one fixed stacking order. This order is weekly booster, daily booster and lastly hourly booster.

A nice feature of the architecture is that it splits the big problem into small problems that can be optimized separately when stacking the forecasters. First, the network performing the *Instant forecaster* should be implemented and optimized. This network architecture can be optimized without the boosting forecasters. Then, the first boosting forecaster is implemented and stacked on top of the *Instant forecaster*. The network architecture of this boosting forecaster can now be optimized separately. Similarly all the network architectures of the other booster forecasters can be optimized one by one. It is still an open issue whether this iterative approach will provide an optimal total network architecture. However, it seems to offer a reasonable good architecture with an easy design flow.

## 3. Case Study

The SBN architecture is applied for forecasting thermal energy consumption of a Finnish office building located in Espoo. The building is a typical office building constructed in 1960s and having four floors.

The main problem is to forecast hourly thermal energy consumption of the building for 24 h in advance. In addition to this main problem, we evaluate the performance in 48 h and 96 h forecasts also. Moreover, the effect of the training data size is evaluated by altering the training data size from 6 months to 6 years. The target metric is Normalized Root Mean Square Error (NRMSE) that is calculated using formula

$$NRMSE = MSE/(\max - \min),$$

where MSE is standard mean square error and maximum and minimum values are calculated over evaluation period.

Available data are the past hourly energy consumption and temperature measurements. Temperature measurements are measured from a weather station being a few kilometres away from the building. For the future timestamps, temperature forecasts are not used in this experiment but real temperature readings from the future timestamps.

### 3.1. Data Set

Thermal energy data set contains hourly energy measurements and hourly temperature measurements from 1.2.2012 to 31.12.2018. The data set is visualized in Figure 8 with an example forecast.

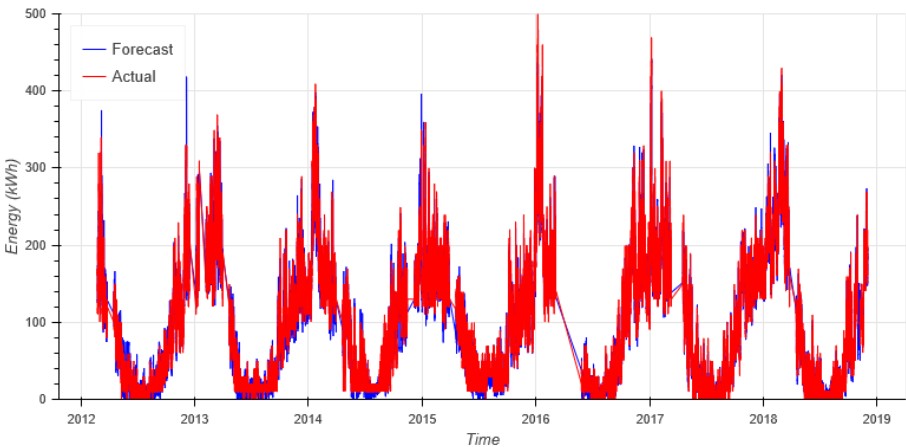

**Figure 8.** The whole data set and 24 h forecasts. Here one can see that there are some missing data that causes direct lines in the figure.

### 3.2. Methodology

The model architecture was optimized by using years 2012–2016 for training and year 2017 for the validation using manual architecture search. When reasonable model architecture was found, the final performance metric was run from year 2018. All the runs that was executed for year 2018 are presented in this paper. After running the metrics for year 2018 no modifications to model parameters was made.

### 3.3. Model

#### 3.3.1. Instant Forecaster

The *Instant forecaster* has three inputs:

1.   Twelve consecutive hours temperature readings before each forecast hour are used.

2. Dummy encoded vector of length 2 determining whether the day is Saturday, Sunday or regular weekday.
3. Predicted hour. Hour is encoded onto unit circle to get continuous variable. Hour is therefore two length vector.

It should be noted that there are many instances of the *Instant forecaster* submodel, and all the instances share the same parameters. Twelve temperature values are first fed through dimensionality reduction submodel that reduces the temperature dimension to one and then all the inputs are processed by two layer densely connected network to get simple forecasts where the hidden layer contains 32 units with dropout regularization.

### 3.3.2. Boosting Forecasters

The problem shall be solved by stacking all the three boosting forecasters in the following order: weekly, daily, hourly. For simplicity all the boosting forecasters shall use the same structure having two fully connected layers where the hidden layer contains 32 units with dropout regularization. The *Weekly boosting forecaster* contains three weeks of data (i.e., two weeks as input and one week to be forecasted). Therefore, we have values $N = 2$ and $k = 1$ in Figure 2. The *Daily boosting forecaster* contains seven days of data indicating values $N = 7$ and $k = 1$ for 24 h forecast in Figure 2. Finally, the *Hourly boosting forecaster* contains 24 h data, indicating values $N = 24$ and $k = 24$ for 24 h forecast. It should be noted that there are multiple instances of each boosting forecaster submodels and the submodels share the same weights. However, for different booster levels, submodel weights are naturally different.

Some statistics of network architectures for each of the used booster setups is shown in Table 1.

**Table 1.** Architectures of different booster combinations.

| Method | Num Layers | Num Parameters |
|---|:---:|:---:|
| Instant forecaster | 3 | 238 |
| Daily booster | 5 | 527 |
| Weekly booster | 5 | 399 |
| Weekly and daily boosters | 7 | 624 |
| Weekly, daily and hourly boosters | 9 | 1457 |

### 3.4. Training Details

The proposed architecture offers multiple different ways of training the network:

- Train the whole network at once using the final prediction value only in loss function
- Train the whole network at once using all the forecasters outputs in loss function but possible with different weights.
- Train first the first prediction submodel and freeze it. Then, train add one boosting forecasters one by one freezing all the earlier forecasters.
- Train iteratively as in previous bullet but do not freeze the earlier forecasters.

We have evaluated all the approaches. The second and the fourth provided roughly equal good results in terms of NRMSE. However, the second one is naturally faster since it needs only one training round. It seems that providing some weights to earlier forecaster loss functions fights well against overfitting and it can be considered as a kind of regularization technique. This technique can also be considered as a kind of a short-cut connection used in ResNets [16,17].

It seems that the training is not very sensitive to the weight given to the *Instant forecaster* and the earlier boosting forecasters. In our case study, we used weight 0.1 for the *Instant forecaster* and earlier boosters and weight 0.9 for the final booster.

We used Adam optimizer [18] in training the network. Besides of learning rate decay of the built-in Adam optimizer, we used an additional exponential learning rate decay. The learning rate decay is dependent of training data size such that for 5 year training data learning rate is reduced by 2% after each epoch. For other training data sizes the decay is altered such that the total decay is same per batch. Initial learning rate is set to 0.0025. Batch size was 256 in our experiments.

Due to learning rate decay and other regularization techniques, early stopping is not needed and the model can be trained with fixed number of epochs.

The network was implemented with Keras library ([19]) using Tensorflow backend ([20]).

### 3.5. Forecast Analysis

This section provides some views of the target data sets and forecasts. First, the whole data set is shown in Figure 8 with an example forecast.

Figure 9 shows the forecasts for our evaluation year 2018. To get better understanding of the forecast and the actual energy consumption, one zoomed snapshot is shown in Figure 10.

For an ideal forecaster the difference between the actual energy consumption and the forecast is a pure white noise. This difference is shown in Figures 11 and 12. From the closer view one can see that the error is correlated but not very much.

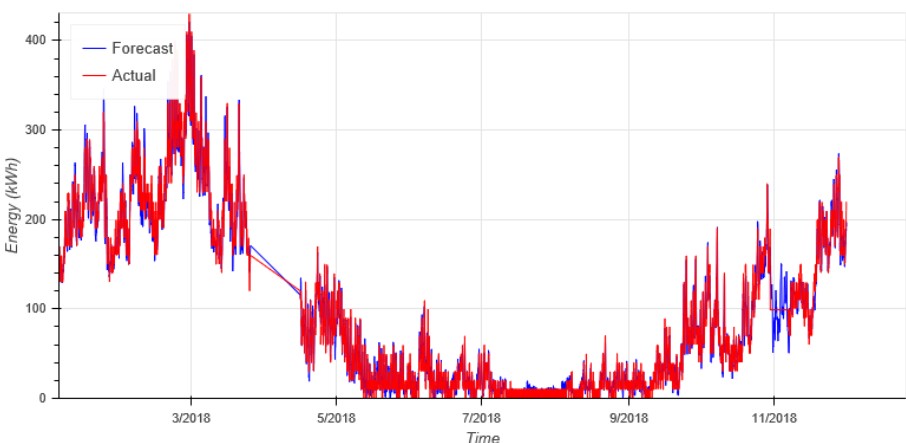

**Figure 9.** 24 h forecasts of the test year 2018.

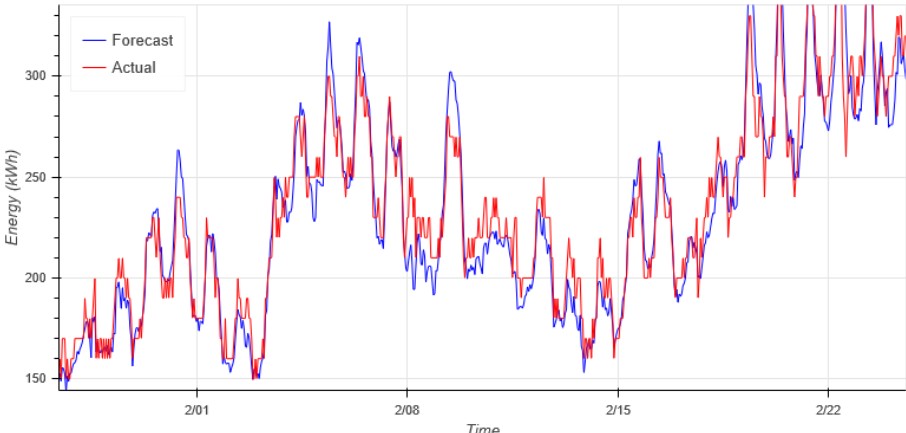

**Figure 10.** 24 h forecasts of a part of the test year 2018.

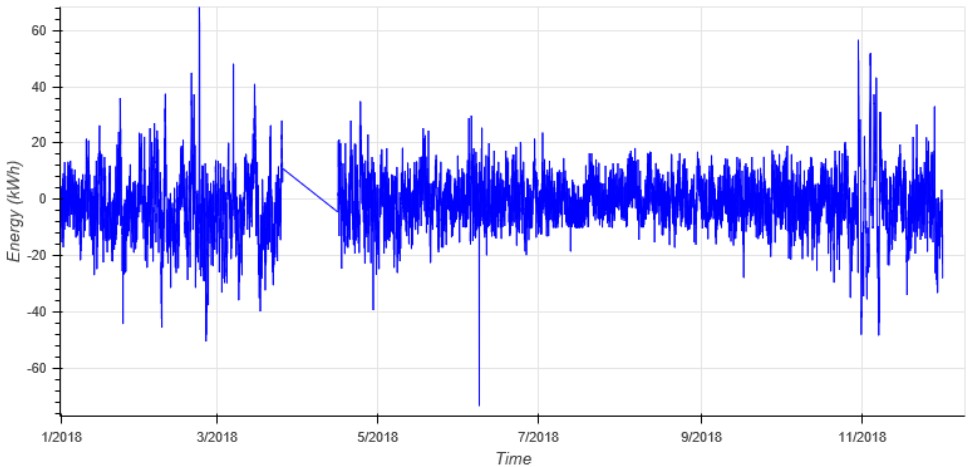

**Figure 11.** Difference of real energy consumption and 24 h forecasted one in the test year 2018.

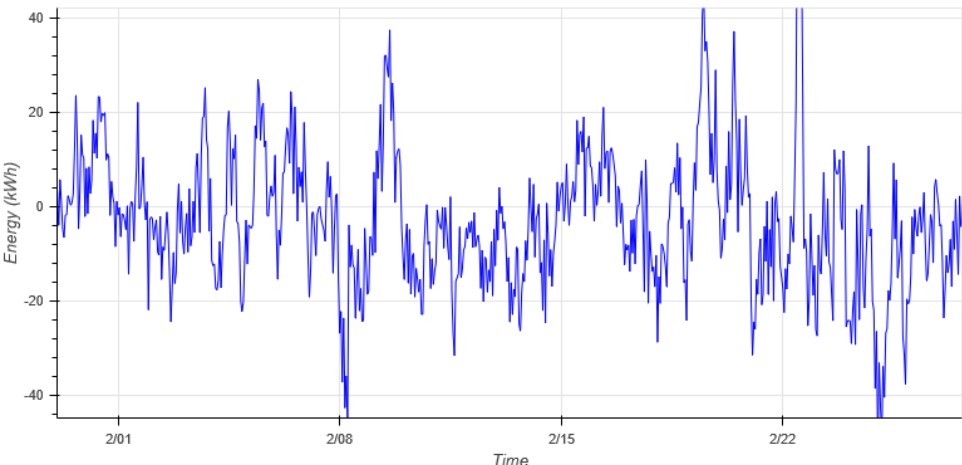

**Figure 12.** Difference of real energy consumption and 24 h forecasted one in a part of year 2018.

### 3.6. Performance Comparison

Performance comparisons of the usage of the different booster setups are presented in Tables 2 and 3, Figures 13 and 14. First, Table 2 and Figure 13 compare the setups by altering the training data size. Here one should note that the basic *Instant forecaster* is the most sensitive to the training data size. When the training data size is only 6 months, and there are no training data available from all temperature conditions, the performance of the *Instant forecaster* decreases a lot. Nevertheless, boosting forecasts are still able to correct these errors to some extent. Moreover, when the training data size increases to 6 years, aging of the data (the building energy consumption has changed over the years so that it does not reflect the situation captured by the older data) starts to decrease the performance of the *Instant forecaster*. Again the boosting forecasters can compensate these errors to some extent. From Table 3 and Figure 14 one should note that usage of the *Hourly boosting forecaster* gets useless when the forecasting period increases since the old hourly energy consumptions do not provide any correlation to future data of different hours. It eventually decreases performance due to overfitting.

**Table 2.** Boosters NRMSE values in percentages in 24 h forecast.

| Method/Training Data Size | 6 Months | 1 Year | 6 Years |
|---|---|---|---|
| Instant forecaster | 6.28 | 3.64 | 4.46 |
| Daily booster | 4.27 | 2.46 | 2.49 |
| Weekly booster | 5.43 | 2.62 | 2.79 |
| Weekly and daily boosters | 4.07 | 2.37 | 2.42 |
| Weekly, daily and hourly boosters | 5.83 | 2.34 | 2.35 |

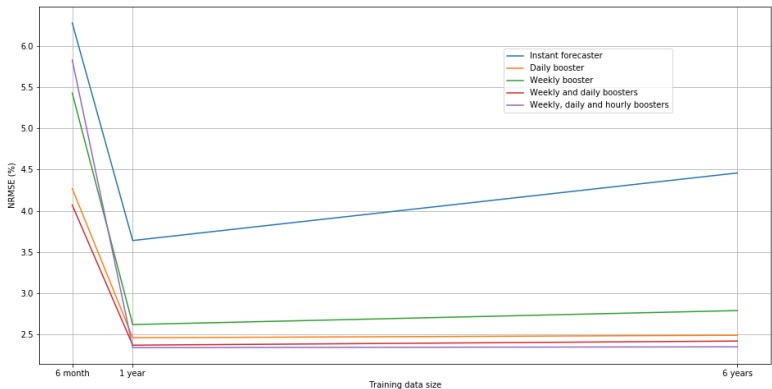

**Figure 13.** NRMSE values of different booster setups in percentages in 24 h forecast.

**Table 3.** Boosters NRMSE values in percentages.

| Method | 24 h Forecast | 48 h Forecast | 96 h Forecast |
|---|---|---|---|
| Instant forecaster | 3.64 | 3.64 | 3.64 |
| Daily booster | 2.46 | 2.62 | 2.64 |
| Weekly booster | 2.62 | 2.62 | 2.62 |
| Weekly and daily boosters | 2.37 | 2.51 | 2.57 |
| Weekly, daily and hourly boosters | 2.34 | 2.51 | 2.66 |

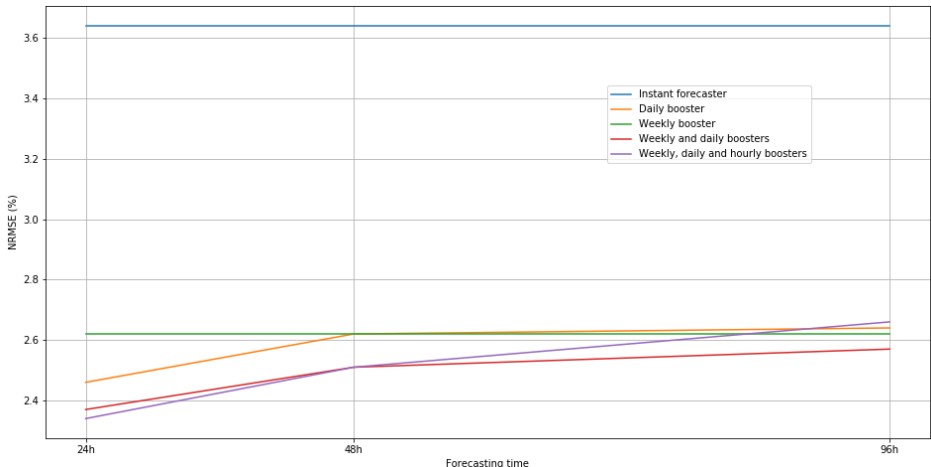

**Figure 14.** NRMSE values of different booster setups for different forecast lengths.

Performance Comparison to the State-of-Art

We used Residual network approach [16] as a presentation of state-of-the-art since it has implementation available (https://github.com/yalickj/load-forecasting-resnet) and it has done good job in public data sets. Let us call this solution simply ResNet. However, the comparison is not completely fair due to following reasons:

- ResNet does not in fact solve the same problem we have stated. The model is suitable for a little bit easier problem where one forecasts energy consumption of each of the hours on the next day given all the energy consumptions of the current day. It utilizes therefore 0–23 more recent observations that we do not utilise depending on the forecast hour.
- The architecture of the SBN is tuned for this particular data set while ResNet is used in plug and play style. However, our network architecture is not very sensitive to changes of each submodel architecture.

The target metrics for comparison is shown in Table 4 and in Figure 15. ResNet solutions seems to perform quite well for plug and play solution when there are a lot of training data available. However, when reducing the amount of training data, it starts to overfit and performance decreases dramatically. However, our SBN solution behaves very well on even a relative small training data size.

**Table 4.** Performance comparison between the SBN and the ResNet. Values are in percentages.

| Training Data | ResNet ([16]) NRMSE | SBN (Ours) NRMSE |
|:---:|:---:|:---:|
| 6 years | 2.57 | 2.35 |
| 5 years | 2.75 | 2.37 |
| 4 years | 3.11 | 2.34 |
| 3 years | 3.51 | 2.40 |
| 2 years | 3.95 | 2.29 |
| 1 years | 13.40 | 2.34 |
| 6 months | 41.91 | 5.83 |

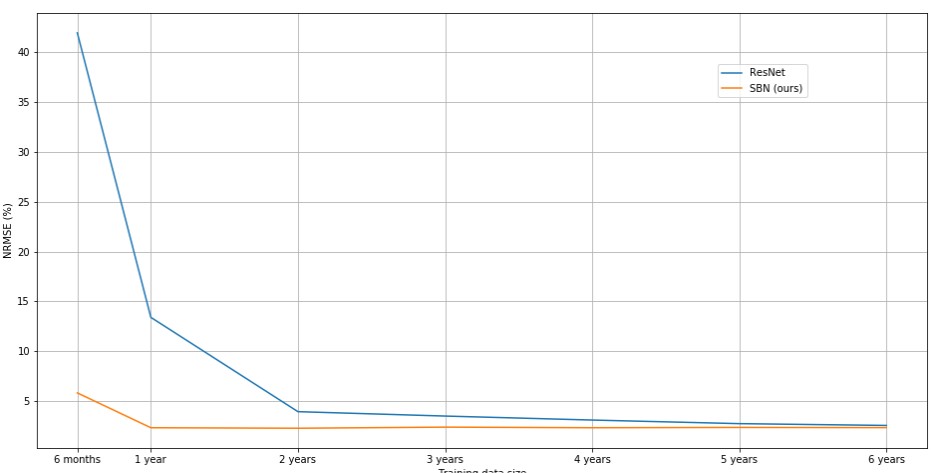

**Figure 15.** NRMSE values of the ResNet ([16]) NRMSE and the SBN(ours) in different training data lengths.

## 4. Conclusions and the Future Work

This paper presented a novel neural network architecture style, called SBN, for short-term energy load forecasting. The SBN aims to achieve the representation capacity of deep learning while at the same time (1) reduce overfitting and (2) provide the capability to adapt to changing data

distributions caused by changes in the modelled energy system. This is achieved with following key ideas, which have driven the design of the SBN: (1) sparse interactions, (2) parameter sharing, (3) equivariant representations, and (4) feedback of residuals from past forecasts. In practice, the SBN architecture consists of a base learner and multiple boosting systems that are modelled as a single deep neural network.

A specific instance of the SBN was implemented on top of Keras and Tersorflow 1.14 to evaluate the architecture style in practice. The evaluation was performed with data from an office building in Finland and the evaluation consisted of following short-term load forecasting tasks: 96 h, 48 h, and 24 h forecasts with 6 years of training data, and 24 h forecast with 1 year and 6 months of training data to evaluate the data-efficiency of the model. As part of the evaluation, the SBN was compared to a state-of-the-art deep learning network [16], which it outperformed in all of the above mentioned forecasting tasks. In particular, the SBN provided significantly better results in terms of data-efficiency.

The SBN is general in the sense that it can be used for other similar time series forecasting domains. As a future work we will apply it to different type of energy forecasting problems and data sets. Moreover, approaches to make the SBN even more data-efficient will be investigated. An interesting idea to this end is related to the model pre-training with transfer learning. As is described in the paper, the SBN is composed of *Instant forecaster* and different boosters. Boosters solve very general univariate time series forecasting problem while Instant forecaster makes the data set specific prediction. Therefore, it could be possible to pre-train the boosters with totally different energy data sets and only train the *Instant forecaster* with the target data set. Since the booster would not need to be trained, this would make the SBN even more data-efficient with respect to training data from the specific building or energy system.

**Author Contributions:** Conceptualization, T.S., J.K. and D.P.; methodology, T.S.; software, T.S.; writing—original draft preparation, T.S. and J.K.; writing—review and editing, T.S., J.K. and D.P.; visualization, T.S.; project administration, D.P.; funding acquisition, D.P. All authors have read and agreed to the published version of the manuscript.

**Funding:** This work has been funded by VTT Technical Research Centre of Finland and Business Finland.

**Conflicts of Interest:** The authors declare no conflict of interest.

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
