# Peer review of "Stacked Boosters Network Architecture for Short-Term Load Forecasting in Buildings"

_energies, doi:10.3390/en13092370_

Round 1

Reviewer 1 Report

The contribution of this paper is not acceptable for this journal. The paper does not read well and requires major revision in some aspects. Furthermore, some parts of the paper should be rewritten and the paper cannot be accepted in the current form. 
1- There is not sufficient theoretical formulation for the proposed control.
2- There is not a clear and identifiable contribution to this study.  3- There are many typos in this paper. 
4- The provided results are not convincing.

Author Response

Point 1: There is not sufficient theoretical formulation for the proposed control. 

Response 1: The paper proposes a novel neural network architecture for building load forecasting and does not include control aspects. 

Point 2: There is not a clear and identifiable contribution to this study

Point 4: The provided results are not convincing.

Response 2 & 4: The main contribution of the paper is a novel neural network architecture, called Stacked Booster Network (SBN), for short-term load forecasting. An instance of the SBN implemented on top of Keras and Tensorflowprovides state-of-the-art results in a case study focusing on office building heating load forecasting. We have improved the paper to describe the SBN, as well as, the case study with results in more detail. 

Point 3: There are many typos in this paper

Response 3: The text has been improved and several typos have been corrected. 

Reviewer 2 Report

The suggested improvements of the text are included in the attached file.

Author Response

Thank you for the valuable comments. The paper has been improved according to your feedback. See details below. 

Line 68: Section numbers should use the Arabic numerals everywhere. Figures 1, 2, 3: 

Corrected

The size of Figure 1 is quite different from that of Figures 2, 3. Consequently the interpretation of indices k and N, coming from Figure 2 (nearly invisible there), needed (at least) in Section 3.3.2, may be not clear.

Corrected

Line 166: The case study has been performed for a “Finish office building” without any detailed specification.  

Some details added

In addition to such specification, the reader may be also interested in more information about computational approach, hardware requirements and software implementation of relevant simulations of building energy consumption. 

SW implementation details added  

Line 170: The following reader-friendly formulation can be recommended: The target metric is the Root Mean Squares Error (RMSE). In addition to RMSE, the Normalized Root Mean Squares Error (NRMSE) should be introduced, too. What method of normalization has been applied for the evaluation of NRMSE in Table 4 ?  

Done. Normalization described

Line 221: The stylistic improvement of the 1st sentence would be welcome. The (formally 1st) footnote refers back to [16] (after slight correction, see lower).

Corrected  

Line 252: The title References is missing. Moreover, all references should be checked carefully to unify their form: e. g. the page range in [11], [13] and [16].  

Corrected. References bib items are copied from original papers.  

Line 281: The correct volume and page range seems to be 2018, 33, 1087– 1088 (not 1 page only). Line 285: The information 26th IEEE International Symposium on Industrial Electronics (ISIE) is missing. Line 292: The correct volume and page range seems to be 2019, 10, 3943– 3952 (not 1 page only). 

Corrected  

Reviewer 3 Report

Good work. Minor English corrections needed, e.g. lines 173-174. I assume that the case study is limited to one building. Is it a typical Finnish office building? Are you able to be more specific about the building used in your case study? Similarly, are you able to be more specific about the source of the dataset you collected for the case study?

As the objective of your case study is to illustrate forecasting of hourly thermal energy consumption of the building for 24h in advance. I think it would useful to show your results in a graph.

Author Response

Minor English corrections needed, e.g. lines 173-174.

Response: Typos and grammar have been improved thorough the paper.  

I assume that the case study is limited to one building. Is it a typical Finnish office building? Are you able to be more specific about the building used in your case study? Similarly, are you able to be more specific about the source of the dataset you collected for the case study?

Response: Yes, case study is limited to one building. We have added some details on the building. 

As the objective of your case study is to illustrate forecasting of hourly thermal energy consumption of the building for 24h in advance. I think it would useful to show your results in a graph.

We have added several graphs to illustrate the results and forecasts in more detail

Reviewer 4 Report

The authors present a novel deep learning architecture for short term load forecasting of building energy loads. The techniques used in the paper are interesting to the readers. However, the presentation of the results needs improvement. 

1) The authors mentioned that "SBN architecture is applied for forecasting thermal energy consumption of Finnish office building", but we don't have other information regarding the building and the building's average thermal energy consumption.

2) The authors used NRMSE and Tables to discuss different setup but it is hard to form a clear picture from those numbers. Visualized graphs can help readers understand the overall performance. 

3) There are some vague descriptions. For example, in section 2.6, it mentions that "the second and the fourth provided equal good results", but what was the definition of good here?

Author Response

A major revision has been done to improve the paper in order to make the contributions and results more clear. 

1) The authors mentioned that "SBN architecture is applied for forecasting thermal energy consumption of Finnish office building", but we don't have other information regarding the building and the building's average thermal energy consumption.

Some details of the office building have been added to the paper. Moreover, three new figures have been added to illustrate the building’s energy consumption with the forecasting results.

2) The authors used NRMSE and Tables to discuss different setup but it is hard to form a clear picture from those numbers. Visualized graphs can help readers understand the overall performance. 

A total of seven new figures have been added to visualize the results. The results section has been improved to make the results more clear. 

3) There are some vague descriptions. For example, in section 2.6, it mentions that "the second and the fourth provided equal good results", but what was the definition of good here?

This sentence has been modified to make it more explicit. 

Reviewer 5 Report

A deeper description of the methodology used for the SBN architecture is missed and the treatment of the error is not found. The model should be properly addressed, defining and ensuring optimal sensitivity analysis for the final solution adopted and a deep description of the working conditions for the forecast energy consumption.

The way authors deal with the training of the network has not been properly addressed and dufficiently explained.

Conclusions section also results confuse for the reader. In this case it should be re-written in order to assess properly the objectives of the work developed here. Please explain it in deep.

The Reviewer.

Author Response

A major revision has been done to improve the paper in order to make the contributions and results more clear. 

Point 1: A deeper description of the methodology used for the SBN architecture is missed and the treatment of the error is not found. The model should be properly addressed, defining and ensuring optimal sensitivity analysis for the final solution adopted and a deep description of the working conditions for the forecast energy consumption.

Response 1: Section 2 introducing the SBN architecture style has been improved to represent the architecture and its main ideas in more detail. In particular, four new figures illustrating the functionality of the boosters have been added. More details on the specific SBN instances implemented for the case study have been also added. Moreover, the results section have been significantly improved (seven new figures illustrating the performance of the SBN).

Point 2: The way authors deal with the training of the network has not been properly addressed and dufficiently explained.

Response 2: Section 3 has been improved with more details on how the SBN has been trained for the particular case study.  

Point 3: Conclusions section also results confuse for the reader. In this case it should be re-written in order to assess properly the objectives of the work developed here. Please explain it in deep.

Response 3: The conclusion section has been completely rewritten. 

Round 2

Reviewer 1 Report

The paper can be accepted in this form. 

Reviewer 4 Report

The authors have made modifications addressing previous comments. I recommend this paper for publication. 

Reviewer 5 Report

The contents regarding my previous concerns have been now properly addressed by Authors.